# Adversarial Music: Real World Audio Adversary Against Wake-word Detection System

**Juncheng B. Li**[1]
junchenl@cs.cmu.edu

**Shuhui Qu**[3]
shuhuiq@stanford.edu

**Xinjian Li**[1]
xinjianl@cs.cmu.edu

**Joseph Szurley**[2]
jszurley@bosch.com

**J. Zico Kolter**[1,2]
zkolter@cs.cmu.edu

**Florian Metze**[1]
fmetze@cs.cmu.edu

[1]Carnegie Mellon University, [2]Bosch Center for Artificial Intelligence, [3]Stanford University

## Abstract

Voice Assistants (VAs) such as Amazon Alexa or Google Assistant rely on *wake-word detection* to respond to people's commands, which could potentially be vulnerable to audio adversarial examples. In this work, we target our attack on the wake-word detection system, jamming the model with some inconspicuous background music to deactivate the VAs while our audio adversary is present. We implemented an emulated wake-word detection system of Amazon Alexa based on recent publications. We validated our models against the real Alexa in terms of wake-word detection accuracy. Then we computed our audio adversaries with consideration of expectation over transform and we implemented our audio adversary with a differentiable synthesizer. Next we verified our audio adversaries digitally on hundreds of samples of utterances collected from the real world. Our experiments show that we can effectively reduce the recognition F1 score of our emulated model from 93.4% to 11.0%. Finally, we tested our audio adversary over the air, and verified it works effectively against Alexa, reducing its F1 score from 92.5% to 11.0%.[1] We also verified that non-adversarial music does not disable Alexa as effectively as our music at the same sound level. To the best of our knowledge, this is the first real-world adversarial attack against a commercial grade VA wake-word detection system.

## 1 Introduction

Adversarial attacks on machine learning systems are a topic of growing importance. As machine learning becomes ever more prevalent in all aspects of modern life, concerns about safety tend to gain prominence as well. Recent demonstrations of the ease with which machine learning systems can be "fooled" have caused a strong impact in the field and in the general media. Systems that use voice and audio such as Amazon Alexa, Google Assistant, Apple Siri and Microsoft Cortana are growing in popularity, and their applications maybe safety critical in cases like in a car. The hidden risk of those advancements is that those systems are potentially vulnerable to adversarial attacks from an ill-intended third-party. Despite the recent growth in consumer presence of audio-based artificial intelligence products, attacks on audio and speech systems have received much less attention than the image and language domains so far.

Despite a number of recent works attempting to create adversarial examples against Automatic Speech Recognition (ASR) systems [Carlini and Wagner, 2018, Schönherr et al., 2018, Qin et al., 2019], we believe robust playable-over-the-air real-time audio adversaries against commercial ASR systems still have not been demonstrated. Attacks that worked digitally against specific models are not effective played over-the-air against commercial ASR models. Moreover, as consumer-facing products, Voice Assistants (VAs) such as Amazon Alexa and Google Assistant are well-maintained by their infrastructure teams. It is revealed that these teams retrain and update ASR models very frequently on their cloud back-end, making a robust audio adversary that can consistently work against these ASR systems almost impossible to craft Hoffmeister [2019]. Attackers would not only lack knowledge of the backend models' parameters and gradients, but would also struggle to keep up with the ever-evolving models.

However, all the existing VAs rely solely on wake-word detection to respond to people's commands, which could potentially make them vulnerable to audio adversarial examples. In this work, rather than directly attacking the general ASR models, we target our attack on the wake-word detection system. Wake-word detection models always have to be stored and executed on-board within smart-home hardware, which is usually very limited in terms of computing power due to form factors. It is also revealed that updates to the wake-word models are much more infrequent and difficult compared to the backend ASR models Hoffmeister [2019]. We therefore propose to attack VAs by "jamming" the wake-word detection, effectively deactivating the VA while our adversary is present. In security term, this is a type of denial-of-service (DoS) attack. Specifically, we create a parametric attack that resembles a piece of background music, making the attack inconspicuous to humans.

We reimplemented the wake-word detection system used in Amazon Alexa based on their latest publications on the architecture [Wu et al., 2018]. We leveraged a large amount of open-sourced speech data to train our wake-word model, testing and making sure it has on par performance compared with the real Alexa. We collected 100 samples of "Alexa" utterances from 10 people and augmented the data set by varying the volume, tempo and speed. We created a synthetic data set using publicly available data sets as background noise and negative speech examples. This collected database is used to validate our emulated model and be compared with the real Alexa.

After successfully training a high-performance emulated wake-word model, we synthesized guitar-like music to craft our audio perturbations. Projected Gradient Descent (PGD) is used here to maximize the effect of our attack while keeping the parameters of our adversarial music within realistic boundaries. During the training of our audio perturbation, we considered expectation over transforms [Athalye et al., 2018] including psychoacoustic masking and room impulse responses. Finally, we tested our perturbation over the air against our emulated model in parallel with the real Amazon Echo and verified that our perturbation works effectively against both of them in a real world setting. Specifically, the recognition F1 score of our emulated model was reduced from 93.4% to 11.0%, and the F1 score of the real Alexa was also brought down from 92.5% to 11.0%. Our adversarial music poses a real threat against commercial grade VAs, leading to potential safety concerns such as distraction in driving and malicious manipulations of smart devices, and thus, our finding calls for future speech research looking into defenses against potential risks and general robustness.

## 2 Background and Related Work

Most current adversarial attacks work by trying to find a way to modify a given input (hopefully by a very small amount) in such a way that the machine learning system's proper functioning is disrupted. A classic example is to take an image classifier and modify an input with a very small perturbation (difficult for human to tell apart from the original image) that still changes the output classification to a completely distinct (and incorrect) one. To achieve such a goal, the general idea behind many of the attack algorithms is to optimize an objective that involves maximizing the likelihood of the intended (incorrect) behavior, while being constrained to a small perturbation. For differentiable systems such as deep networks, which are the current state-of-the-art for many classification tasks, utilizing gradient-based methods is a common approach. We describe such methods and their relation to our work in more depth in Section. 3.5. In this work, our target of attack is wake-word detection systems.

Adversarial attacks were initially introduced for images [Szegedy et al., 2013] and have been studied the most in the domain of computer vision [Nguyen et al., 2015, Kurakin et al., 2016, Moosavi-Dezfooli et al., 2016, Elsayed et al., 2018, Li et al., 2019]. Following successful demonstrations in the vision domain, adversarial attacks were also successfully applied to natural language processing [Papernot et al., 2016, Ebrahimi et al., 2018, Reddy and Knight, 2016, Iyyer et al., 2018, Naik et al., 2018]. This trend gives rise to defensive systems such as Cisse et al. [2017], Wong and Kolter [2018], and thus provides a guideline to the community about how to build robust machine learning models.

Attacks on audio and speech systems have received much less attention so far. Two years ago, Zhang et al. [2017] pioneered a proof-of-concept that proved the feasibility of real-world attacks on speech recognition models. This work however, had a larger focus on the hardware part of the Automatic Speech Recognition (ASR) system, instead of its machine learning component. Until very recently, there was not much work done on exploring adversarial perturbation on speech recognition models. Carlini et al. [2016] was the first to demonstrate that attack against HMM models are possible. They claimed to effectively attack based on the inversion of feature extractions. This work was preliminary since it only showcased a limited number of discrete voice commands, and the majority of perturbations are not able to be played over the air. As a follow-up work, Carlini and Wagner [2018], Qin et al. [2019] showcased that curated white-box attacks based on adversarial perturbation can easily fool the Mozilla speech recognition system. Again, their attacks would only work in with their special setups and are very brittle in the real world. Meanwhile, Schönherr et al. [2018] attempted to leverage psycho-acoustic hiding to improve the chance of success of playable attacks. They verified their attacks against the Kaldi ASR system, whereas the real-world success rate was still not satisfying, and the adversary itself cannot be played from a different source. Unfortunately, state-of-the-art audio adversaries can only work digitally against the specific model they were trained against, but cannot work robustly over the air against state-of-the-art commercial grade ASR systems. We verified all these adversaries mentioned above not effective against Alexa or Siri. Alexa and Siri can still transcribe correctly with the presence of these audio perturbations. Acknowledging the strict limit on the potency of such an over-the-air attack, we aim at a different target rather than focusing on the ASR models deployed in commercial products. Our proposed attack targets at the more manageable but equally critical wake-word detection system, and effectively demonstrates that it can be playable over the air.

*Here are our main contributions in this work compared with previous works:*

1. *We create a parametric threat model in audio domain that allows us to disguise our adversarial attack as a piece of music playable over the air in the physical space.*

2. *Our adversarial attack is a "gray-box" attack[2] that leverages the domain transferability of our perturbation. This is a lot more challenging than previous works on white-box attack where attackers have perfect information about the model.*

3. *Our adversarial attack is jointly optimizing the attack nature while fitting the threat model to the perturbation achievable by the microphone hearing response of Amazon Alexa. Our attack budget is very limited compared with previous works, which makes this challenging.*

4. *Our adversarial attack demonstrated its effect in the real world under separate audio source settings, which is the first real-time "gray-box" adversarial attack against commercial grade Voice Assist's wake-word detection system to our knowledge. [3]*

## 3 Synthesizing Adversarial Music against Alexa

This section contains the main methodological contributions of our paper: the algorithmic and practical pipeline for synthesizing our adversarial music. To begin, we will first describe the training setup and the performance of our emulated wake-word detection model. We then describe the general threat model we consider in this work. Unlike past works which often considered $L_p$ norm bounded perturbations on the entire spectrum, we require a threat model that can generate a piece of audible music. Next, we describe the approach we used to make our threat model robust over the air. Finally,

we describe how we optimized the parameters of our threat model to synthesize a robust over-the-air attack.

## 3.1 Emulate the Wake-word Detection Model

Wake-word detection is the first important step before any interactions with distant speech recognition. Due to the compacted space of embedded platforms and the need for quick reflection time, models of wake-word detection are usually truncated and vulnerable to attacks. Thus, we target our attack at the wake-word detection function.

Since the Alexa model is a black-box to us, the only way to attack it is either to estimate its gradients, or to emulate its architecture and later transfer the white-box attack against the emulated model to its original model. Estimating its gradient using first order optimization techniques would be extremely computationally expensive [Ilyas et al., 2018], making it difficult if not impossible to implement. Luckily, the architecture of Amazon Alexa was published in Panchapagesan et al. [2016], Kumatani et al. [2017], Guo et al. [2018], allowing us to emulate the model as if it is a white-box attack. We implemented the time-delayed bottleneck highway networks with Discrete Fourier Transform (DFT) features as shown in Figure. 1. The architecture is following the details in Guo et al. [2018], which is the most up-to-date information on the model architecture.

The architecture of the emulated model is shown in Figure. 1. The model is a two-level architecture which uses the highway block as the basic building block. Specifically, our architecture contains a 4-layer highway block as a feature extractor, a linear layer acting as the bottleneck, a temporal context window that concatenates features from adjacent frames, and a 6-layer highway block for classification.

The training data for wake-word detection systems is very limited, so our model is first pre-trained with several large corpora [Cieri et al., 2004, Godfrey et al., 1992, Rousseau et al., 2012] to train a general acoustic model. Then it is adapted to the wake-word detection model by using a small amount of wake-word detection training data. The emulated model could detect the wake-word "Alexa" by recognizing the corresponding phonemes of AX, L, EH, K, S and AX.

We trained the model as a binary classification problem over a time sequence, distinguishing between wake-words and non wake-words. The performance is evaluated over a reserved test set. Care has been taken to ensure that augmented copies of the same raw audio sample will not occur in the train and test set simultaneously. Common performance metrics are listed in Table 1. Given that our emulated model's architecture is very similar to the Alexa model, the gradients of the two models should also share great similarities. We believe that the white-box attack computed by gradient based attacks against our emulated model would be effective against Alexa's original model. Figure. 3.1 shows the Detection Error Tradeoff (DET) in the similar style as Guo et al. [2018]. We note that our performance is not exactly the same as Alexa model due to the lack of training data, but our performance is in the same order of magnitude. Thus, we believe we should have a high fidelity emulation of the Alexa wake-word detection model.

## 3.2 Adversarial Music Synthesizer (Threat Model)

To perform the adversarial attack on the audio domain, we introduce a parametric model to define a realistic construction of our adversary $\delta_\theta$ parameterized by $\theta$. We use the Karplus-Strong algorithm Jaffe and Smith [1983] to synthesis guitar-timbre sounds. The goal is to generate a sequence of $L$ guitar notes $\{(fr_i, d_i, vol_i)\}_{i=1}^{L}$, with given Beats Per Minute (BPM) $bpm$, Sampling Frequency $fr_s$, where $fr_i$ is the frequency pitch of the $i_{th}$ note, $d_i$ is the note duration, and $vol_i$ is the note volume. In this work, we update the $\theta = \{(fr_i, vol_i)\}_{i=1}^{L}$ of all notes' frequency $fr$ and volume $vol$, while fixing the notes' duration $\{d_i\}_{i=1}^{L}$.

The Karplus-Strong algorithm is an example of digital waveguide synthesis to simulate string instruments physically. It mimics the dampening effects of a real guitar string as it vibrates by taking the decaying average of two consecutive samples: $y[n] = \gamma(y[n - D] + y[n - D + 1])/2$, where $\gamma$ is the decay factor, $y$ is the output wave, $D$ is the sampling phase delay. This is similar to a one-zero low-pass filter with a transfer function $H(z) = (1 + z^{-1})/2$ [Smith, 1983], as shown in Figure. 3.

When a guitar string is plucked, the string vibrates and emits sound. To simulate this, the Karplus-Strong algorithm consists of two phases, as shown in Figure. 3:

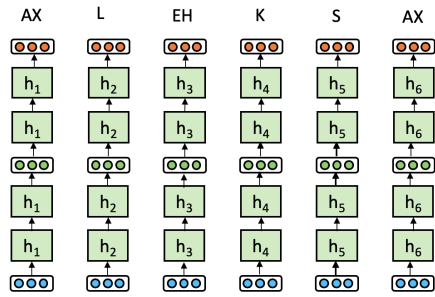

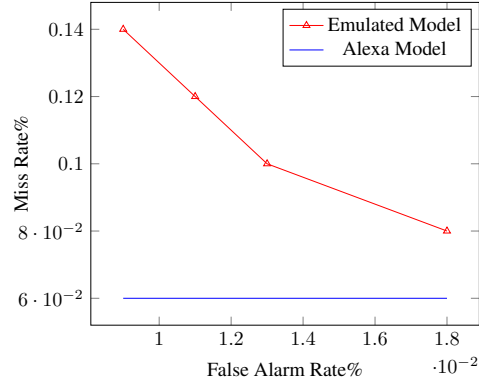

Figure 1: Emulated Wake-word model

Figure 2: Detection Error Tradeoff Curve. The curve of Alexa model is shown in a flat line as its False Alarm Rate is not published

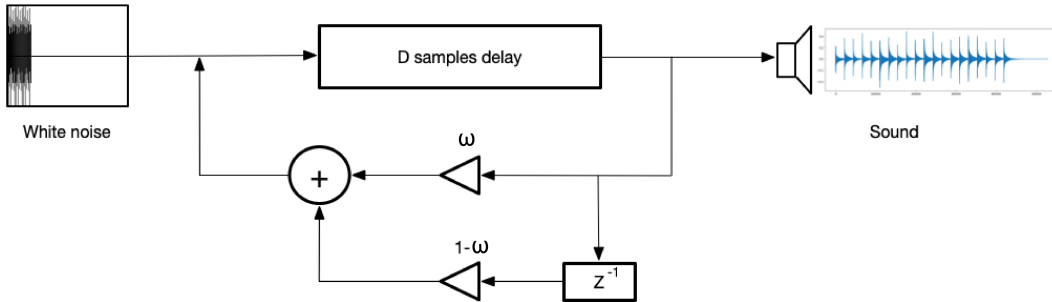

Figure 3: String instruments with the one-zero low-pass filter approximation. The synthesis process first generates a short excitation $D$-length waveform. It is then fed into the filter iteratively to generate the sound.

**Plucking the string**: The string is "plucked" by a random initial displacement and initial velocity distribution $y[0 : D - 1] \sim \mathcal{N}(0, \beta)$, where $\beta$ is the displacement factor. The plucking time, i.e. the sampling phase delay $D$ for the note frequency $fr$ is calculated using $D = fr_s/fr$.

**The resulting vibrations**: The pluck causes a wave-like displacement over time with a decaying sound. The decay factor depends on the note frequency $fr$ and the delay period $n_D = d \times fr/fr_s$. It is calculated as: $\gamma = (4/log(fr))^{1/n_D}$. The volume of the output of a note is adjusted by $v_{output} = p_v \times vol$ using a frequency-specific volume factor $p_v = 1 + 0.8 \times (log(fr) - 3)/5.5 \times cos(\pi/5.3(log(fr) - 3))$ [Woodhouse, 2004]. Jaffe and Smith [1983] suggested using a linear interpolation to obtain a fractional delay to generate a better result:

$$y[n] = \gamma v_{output}(\omega \times y[n - D] + (1 - \omega) \times y[n - D - 1])/2, \tag{1}$$

where, $\omega \in (0, 1)$ is the weight factor. Overall, the Karplus-Strong algorithm is shown in algorithm. 1.

---

**Algorithm 1:** Karplus-Strong algorithm

---

**1** Simulate the plucking phase of each note $i$ by initialize $y_i[0 : D - 1] \sim \mathcal{N}(0, \beta)$;
**2 for** $i = 1, ..., L$ **do**
    **for** $n = D, ..., d_i$ **do**
        $y_i[n] = \gamma v_{output}(\omega \times y_i[n - D] + (1 - \omega) \times y_i[n - D - 1])/2$;
**3** Return $y$;

---

The resulting synthesizing function $\pi(x; \theta)$ is a differentiable function of part of the parameters $\theta$ value is a continuous function of the parameters: the frequency $f$ and the volume $vol$ of each note (We fix the note duration $d$). Therefore, we can implement the perturbation model within an automatic differentiation toolkit (we implement it with the PyTorch library), a feature that will be exploited to both fit the parametric perturbation model to real data, and to construct real-world adversarial attacks.

## 3.3 Psychoacoustic Effect

Our ultimate task is to deceive the voice assistant with minimal impact to human hearing, and it is natural to leverage the psychoacoustic effect of human hearing. The principles of the psychoacoustic model are similar to what used in the compression process of audio files, e.g. compress the lossless file format "wav" to the lossy file format "mp3". In this process, the information carried by the audio file is actually altered while human ear could not tell the differences between these two sounds. Specifically, a louder signal (the "masker") can make other signals at nearby frequencies (the "maskees") imperceptible [Mitchell, 2004]. In this work, we adopted the same setup as Qin et al. [2019]. When we add an perturbation $\delta$, the normalized power spectral density (PSD) estimate of the perturbation $\bar{p}_\delta(k)$ is under the frequency masking threshold of the original audio $\eta_x(k)$,

$$\bar{p}_\delta(k) = 92 - \max_k p_x(k) + p_\delta(k) \tag{2}$$

where $p_\delta(k) = 10 \log_{10} |\frac{1}{N} s_\delta(k)|^2$, $p_x(k) = 10 \log_{10} |\frac{1}{N} s_x(k)|^2$ are power spectral density estimation of the perturbation and the original audio input. $s_x(k)$ is the $k_{th}$ bin of the spectrum of frame $x$. This results in the loss function term:

$$L_\eta(x, \delta) = \frac{1}{\lfloor \frac{N}{2} \rfloor + 1} \sum_{k=0}^{\lfloor \frac{N}{2} \rfloor} \max\{\bar{p}_\delta(k) - \eta_x(k), 0\} \tag{3}$$

where $N$ is the predefined window size and $\lfloor x \rfloor$ outputs the greatest integer no larger than $x$.

## 3.4 Expectation Over Transform

When using the voice assistant in a room, the real sound caught by the microphone includes both the original sound spoken by human and the reflected sound. The "room impulse response" function explained the transform of the original audio and the audio caught by the microphone. Therefore, to make our adversarial attack effective in the physical domain, i.e. attack the voice assistant over the air, it is necessary to consider the room impulse response in our work. Here, we use the classic Image Source Method introduced in Allen and Berkley [1979], Scheibler et al. [2018] to create the room impulse response $r$ based on the room configurations (dimension, source location, reverberation time): $t(x) = x * r$, where $x$ denotes clean audio and $*$ denotes convolution operation. The transformation function $t$ follows a chosen distribution $\mathcal{T}$ over different room configurations.

## 3.5 Projected Gradient Descent and Loss Formulation

Originally, wake-word detection problem is formulated as a minimization of $\mathbf{E}_{x,y \sim D}[L(f(x), y)]$ where $L$ is the loss function, $f$ is the classifier mapping from input $x$ to label $y$, and $D$ is the data distribution. We evaluate the quality of our classifier based on the loss, and a smaller loss usually indicates a better classifier. In this work, since we are interested in attack, we form $\max_\delta[\mathbf{E}_{x,y \sim D}[L(f(x'), y)]]$, where $x' = x + \delta$ is our perturbed audio. In a completely differentiable system, an immediately obvious initial approach to this would be to use gradient ascent in order to search for an $x'$ that maximizes this loss. However, for this maximization to be interesting both practically and theoretically, we need $x'$ to be close to the original datapoint $x$, according to some measure. It is thus common to define a perturbation set $C(x)$ that constrains $x'$, such that the maximization problem becomes $\max_{x' \in C(x)}[\mathbf{E}_{x,y \sim D}[L(f(x'), y)]]$. The set $C(x)$ is usually defined as a ball of small radius (of either $\ell_\infty, \ell_2$ or $\ell_1$) around $x$.

Since we have to solve such a constrained optimization problem, we cannot simply apply the gradient descent method to maximize the loss, as this could take us out of the constrained region. One of the most common methods utilized to circumvent this issue is called Projected Gradient Descent (PGD).

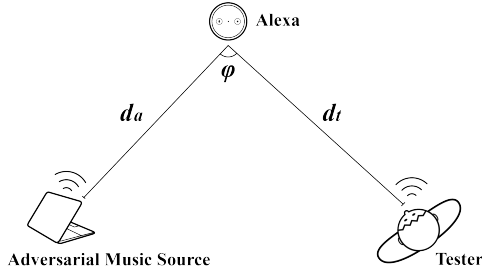

Figure 4: Physical Testing Illustration

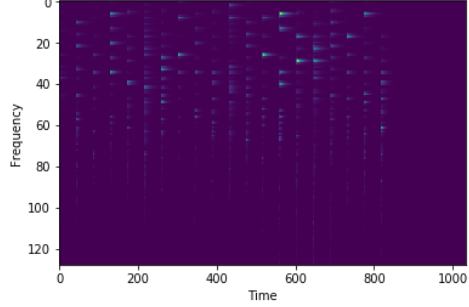

Figure 5: A sample adversarial music

To actually implement gradient descent methods, instead of maximizing the aforementioned loss $L$, we will invert the sign and minimize the negative loss, *i.e.*, $\min\limits_{x' \in C(x)} [-l(x, \delta, y)]$.

In our case, we attack the voice assistant via a parametric threat model and considering psychoacoustic effect, as illustrated in Section. 3.3 and Section. 3.2. The loss function of attack thus can be rewritten as:

$$\max l(x, \delta_\theta, y) = \mathbf{E}_{x,y \sim D}[L_{\text{wake}}(f(x + \delta_\theta), y) - \alpha \cdot L_\eta(x, \delta_\theta)] \tag{4}$$

where $L_{\text{wake}}$ is the original loss of the emulated model for wake-word detection, and $\alpha$ is the balancing parameter. $\delta_\theta$ is our adversarial music defined in Section. 3.2. In addition, we consider the room impulse response defined in Section. 3.4, which would be the attack in physical world over the air. We simulated our testing environments using the parameters shown in Table 2 to transform $t(x)$ our digital adversary to compensate for the room impulse responses. We want to optimize our $\delta$ by tuning $\theta$ to maximize this loss to synthesize our adversary:

$$\max l(x, \delta_\theta, y) = \mathbf{E}_{t \in \mathcal{T}, x,y \sim D}[L_{\text{wake}}(f(t(x + \delta_\theta), y)) - \alpha \cdot L_\eta(x, \delta_\theta)] \tag{5}$$

## 4 Experiments and Results

### Datasets

We collected 100 positive speech samples (speaking "Alexa") from 10 peoples (4 males and 6 females; 4 native speakers of English, 6 non-native speakers of English). Each person provided 10 utterances, under the requirement of varying their tone and pitch as much as possible. We further augmented the data to 20x by varying the speed, tempo and the volume of the utterance, resulting in 2000 samples. We used LJ speech dataset [Ito, 2017] for background noise and negative speech examples (speak anything but "Alexa"). We created a synthetic data set by randomly adding positive and negative speech examples onto a 10s background noise and created binary labels accordingly. While "hearing" positive speech examples, we set label values as 1.

### Training and Testing the Adversarial Music

We followed the methodology described in Section. 3.5, using the loss function defined by Eq. 5 and the parametric method illustrated in Section. 3.2, we optimized the parameters of the music to maximize our attack. We used the emulated model developed in Section. 3.1 to estimate the gradient and maximizes the classification loss following Eq. 5. When using PGD to train, we restricted the frequency to 27.5Hz $\sim$ 4186Hz in the 88 notes space, and restricted the volume from 0 dBA $\sim$ 100 dBA. Other parameters are defined in the code, we fixed some parameters to speed up the training. The trained perturbations are directly added on top of the clean signals to perform our digital evaluation. Our physical experiments are conducted at a home environment which can be seen in the video attached in the supplementary material, and the setup is shown in Figure. 4. The adversarial music is played by a MacBook Pro (15-inch, 2018) speaker[4] The tester stands $d_t$ away

from the Amazon Echo or the computer running our emulated model, and the adversary is placed $d_a$ away from the Echo. The angle between the two lines is $\varphi$. We tested against the model for 100 samples with and without the audio adversary. We used a decibel meter to ensure that our adversary is never louder than the wake-word command. Our human spoken wake-word command is measured to be 70 dBA on average. We experimented with adversaries being played at from 40 dBA and 70 dBA (measured by the decibel meter) with the background reference noise at 20 dBA, and we also experimented with 2 different testers (a male and a female). We also experimented with the influence of timing offset $t_{offset}$ on the performance of the adversarial music. [5]

| Models | Attack | Digital / Physical | Precision | Recall | F1 Score | # Samples |
|---|---|---|---|---|---|---|
| Emulated Model | No | Digital | 0.97 | 0.94 | 0.955 | 4000 |
| Emulated Model | No | Physical | 0.96 | 0.91 | 0.934 | 100 |
| Alexa | No | Physical | 0.93 | 0.92 | 0.925 | 100 |
| Emulated Model | Yes | Digital | 0.14 | 0.11 | 0.117 | 4000 |
| Emulated Model | Yes | Physical | 0.12 | 0.09 | 0.110 | 100 |
| Alexa | Yes | Physical | 0.11 | 0.10 | 0.110 | 100 |

Table 1: Performance of the models with and without attacks in digital and physical testing environments given the number of testing samples

The performance metrics of the emulated model on adversary examples are shown in Table 1. An example of modified adversarial attack example is shown in Figure .5. As we can see, our attack is effective against both the emulated model and the real Alexa model in both digital and physical tests. Especially, the precision score takes a heavier hit by the adversary compared with the recall score. We also noticed that False Positives are relatively uncommon, this might be due to the fact that we are running an untargeted attack against the wake-word detection, and our adversaries are not encouraging the model to predict a specific target label. Overall, our experiments showcased that audio adversaries masked as a piece of music can be played over the air, and disable a commercial grade wake-word detection system.

| Test Against Alexa | $\varphi = 0°$ | | | $\varphi = 90°$ | | | $\varphi = 180°$ | | |
|---|---|---|---|---|---|---|---|---|---|
| $d_t =$ | $4.2ft$ | $7.2ft$ | $10.2ft$ | $4.2ft$ | $7.2ft$ | $10.2ft$ | $4.2ft$ | $7.2ft$ | $10.2ft$ |
| $d_a = 4.7ft, 70dBA$ | 0/10 | 0/10 | 0/10 | 0/10 | 0/10 | 0/10 | 0/10 | 0/10 | 0/10 |
| $d_a = 6.2ft, 70dBA$ | 1/10 | 0/10 | 0/10 | 1/10 | 0/10 | 0/10 | 1/10 | 2/10 | 1/10 |
| $d_a = 7.7ft, 70dBA$ | 2/10 | 0/10 | 0/10 | 3/10 | 1/10 | 1/10 | 3/10 | 3/10 | 1/10 |
| $d_a = 4.7ft, 60dBA$ | 0/10 | 0/10 | 0/10 | 0/10 | 0/10 | 0/10 | 0/10 | 0/10 | 0/10 |
| $d_a = 6.2ft, 60dBA$ | 1/10 | 1/10 | 0/10 | 3/10 | 1/10 | 0/10 | 2/10 | 2/10 | 0/10 |
| $d_a = 7.7ft, 60dBA$ | 2/10 | 1/10 | 0/10 | 3/10 | 2/10 | 1/10 | 4/10 | 3/10 | 1/10 |
| $d_a = 4.7ft, 50dBA$ | 1/10 | 2/10 | 1/10 | 2/10 | 2/10 | 2/10 | 2/10 | 2/10 | 1/10 |
| $d_a = 6.2ft, 50dBA$ | 2/10 | 3/10 | 2/10 | 3/10 | 3/10 | 2/10 | 2/10 | 3/10 | 2/10 |
| $d_a = 7.7ft, 50dBA$ | 2/10 | 3/10 | 2/10 | 3/10 | 2/10 | 3/10 | 4/10 | 3/10 | 3/10 |
| $d_a = 4.7ft, 40dBA$ | 3/10 | 4/10 | 3/10 | 4/10 | 3/10 | 4/10 | 4/10 | 4/10 | 4/10 |
| $d_a = 6.2ft, 40dBA$ | 3/10 | 4/10 | 4/10 | 4/10 | 4/10 | 4/10 | 4/10 | 4/10 | 4/10 |
| $d_a = 7.7ft, 40dBA$ | 3/10 | 4/10 | 4/10 | 5/10 | 5/10 | 4/10 | 5/10 | 4/10 | 5/10 |

Table 2: Times of the real Amazon Alexa being able to respond to the wake-word under the influence of our adversarial music with different settings. (The female and male tester each tests 5 utterances.) The testing set up is illustrated in Figure. 4, and it is also showed in the demo video.

In our physical experiments against the real Alexa, we measured the performance of our audio perturbation under several different physical settings shown in Table 2. As we can observe, our model successfully attacked against Alexa in most cases. The distance $d_a$ between Alexa and the attacker is critical for a successful attack: a shorter distance $d_a$ generally leads to a more effective adversary, while the number of successful attacks declines when distance $d_t$ is shorter. The adversary effectively fooled Alexa at most volume levels. The success rate increased as the volume of the adversarial music increased from 40 dBA to 70 dBA. We also observed it would not be effective

being played under 40 dBA. We also experimented with different starting time of the adversary and wake words. As is shown in Table 3, the adversary was effective when it overlapped with the entire length of the wake-word, while it was ineffective if the wake-word uttered first. Our adversaries successfully demonstrated to be effective at various physical locations and angles $\varphi$ referenced to Alexa and the human tester, the larger $\varphi$ the less effective our adversary would be. This is relevant because Alexa's 7-microphone array is supposedly to be very robust at source separation. This can prove that our consideration of the room impulse responses was useful. When we remove the room impulse transform, the adversary lost its effect.

To verify that Alexa would not be affected easily with other non-adversarial noises, we experimented with three different baselines: random music generated by the Karplus-Strong (KS) algorithm, single-note sounds generated by the KS algorithm and random pieces of music.[6] (not generated by the KS algorithm). The baseline noises are played at the same volume as our adversarial music. As is shown in Table 4, these three baselines could rarely fool Alexa under various testing conditions. This demonstrated the Projected Gradient Descent (PGD) adversarial music trained using the KS algorithm is effective.

| Test Against Alexa | $\varphi = 0°$ | | | $\varphi = 90°$ | | | $\varphi = 180°$ | | |
|---|---|---|---|---|---|---|---|---|---|
| $d_t =$ | $4.2ft$ | $7.2ft$ | $10.2ft$ | $4.2ft$ | $7.2ft$ | $10.2ft$ | $4.2ft$ | $7.2ft$ | $10.2ft$ |
| $t_{offset} = +1s$ | 10/10 | 10/10 | 10/10 | 10/10 | 10/10 | 10/10 | 10/10 | 10/10 | 10/10 |
| $t_{offset} = -1s$ | 0/10 | 0/10 | 0/10 | 0/10 | 0/10 | 0/10 | 0/10 | 0/10 | 0/10 |
| $loop$ | 0/10 | 1/10 | 0/10 | 0/10 | 1/10 | 0/10 | 1/10 | 0/10 | 1/10 |

Table 3: Times of the real Amazon Alexa being able to respond to the wake-word under the influence of our adversarial music with different time-offset. The adversarial music was placed at $d_a = 7.7ft$ away from the Echo, and played at $70dBA$. Here, $t_{offset}$ is +1 indicates the adversarial music is played 1 second after the wake word; $loop$ indicates non-stop adversarial music.

| Test Against Alexa | $\varphi = 0°$ | | | $\varphi = 90°$ | | | $\varphi = 180°$ | | |
|---|---|---|---|---|---|---|---|---|---|
| $d_t =$ | $4.2ft$ | $7.2ft$ | $10.2ft$ | $4.2ft$ | $7.2ft$ | $10.2ft$ | $4.2ft$ | $7.2ft$ | $10.2ft$ |
| Random Music | 10/10 | 10/10 | 10/10 | 10/10 | 9/10 | 10/10 | 10/10 | 10/10 | 10/10 |
| Random Notes | 9/10 | 10/10 | 10/10 | 9/10 | 10/10 | 10/10 | 10/10 | 9/10 | 9/10 |
| Real Music | 10/10 | 10/10 | 10/10 | 10/10 | 10/10 | 10/10 | 10/10 | 10/10 | 9/10 |

Table 4: Times of the real Amazon Alexa being able to respond to the wake-word under the influence of baseline noises: Random music and random single notes generated by the Karplus-Strong (KS) algorithm, and real guitar music which are not generated by the KS algorithm. The adversarial music was placed at $d_a = 7.7ft$ away from the Echo, and played at $70dBA$.

## 5  Conclusion

In this work, we demonstrated a DoS attack on the wake-word detection system of Amazon Alexa with a real-word inconspicuous adversarial background music. We first created an emulated model for the wake-word detection on Amazon Alexa following the implementation details published in Guo et al. [2018]. The model is the basis to design our "gray-box" attacks. We collected, augmented and synthesized a data set for training and testing. We implemented our threat model which disguises our attack in a sequence of inconspicuous background music notes in PyTorch. Our experiments verified that our adversarial music is not only effective against our emulated model digitally, but also can effectively disable Amazon Alexa's wake-word detection function over the air. We also verified that non-adversarial music does not affect Alexa compared to our adversary at the same sound level. Our work shows the potential of adversarial attack in the audio domain, specifically, against the widely applied wake-word detection systems in voice assistants. Overall, this suggests a real concern of attack against commercial grade machine learning algorithms, highlighting the importance of adversarial robustness from a practical, security-based point of view.

**Acknowledgement**: The authors would like to thank Bingqing Chen and Zhuoran Zhang for their help with data collection and preliminary studies.

## Footnotes

[1]Our code and demo videos can be accessed at https://www.junchengbillyli.com/AdversarialMusic; F1 score is the metric we chose here since detection and false alarms are equally damaging

[2]We avoid using the term "black-box" here since we admit that the gradients of the real Alexa model can be expected to be similar to those of models of Panchapagesan et al. [2016]

[3]Our demo video is included in the supplementary material

[4]We have also experimented with Alienware MX 18R2 speakers and Logitech Z506 speakers, MacBook Pro speakers generated the best results. Since we are not comparing different speakers in this paper, we report the results from the MacBook Pro speakers.

[5] please refer to the demo video in https://www.junchengbillyli.com/AdversarialMusic

[6]We picked 10 different top ranking guitar music on Youtube.

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
