[Reviews · NeurIPS 2019]

Reviewer 1



Overall, I found this paper to be interesting, clearly written, and it describes an effective denial-of-service (DoS) attack for voice assistant systems. Topically, it's worth noting that DoS for this kind of system may be a rather "low-stakes" attack, since it (by definition) cannot be used to violate user privacy or execute commands without authorization. However, it may be a technically more interesting attack than a false-positive triggering attack, which could be trivially achieved by playing back the wake-word. The fact that a DoS attack is possible with relatively innocuous-sounding noise ought to be broadly interesting to audio and/or security researchers, though I'm not sure how well that fits the NeurIPS audience. Quality-wise, the authors have done a thorough job of reimplementing a known, but proprietary model, and evaluating the efficacy of their method in both simulation and physical instantiations. There are some minor technical errors in the description of the method, but no show-stoppers that I can tell. The paper is generally clear, though it could benefit from a bit more focus in setting up the problem. The abstract suggests a focus on false-negative (DoS) errors, but this seems to be post-hoc reasoning after trying to generate adversarial noise for two-sided error. It's not surprising that false positives were difficult (impossible?) to achieve with their noise model, and the presentation could be clarified a bit up front to maintain the focus on false-negatives throughout the derivation. Minor comments: - Line 43, "It is also revealed that updates..." this claim needs more support or a reference. - Line 50, "on-pair" -> "on par" - Line 165, citation needed for Karplus-Strong - Line 168, "frequency (key) of the ith note" -- incorrect use of "key" here, musically speaking. "Pitch" would be more appropriate. - Line 223, since the noise pattern is fixed across all examples, shouldn't the maximum be outside the expectation? - Line 236, 240, eq 4 & 5: it's unclear how x, x', z, and delta are related in this equation. Wake-word loss should depend on y.

Reviewer 2



This paper ultimately describes a black box adversarial attack on the wake-word portion of the Amazon Echo, whereby playing a particular melody on a plucked string model can interfere with the Echo's ability to detect the wake-word "Alexa". Real guitar music played at the same dB SPL level hardly interferes at all. In order to achieve this, the attack was developed on an implementation of the Alexa wake-word detector described in the literature. Reverberation was simulated in this attack as well. It was then transferred to the real Echo system in the real world. Experiments were carried out using 100 recordings from 10 talkers in various configurations relative to the device and the jammer signal, showing that the detection F1 score was reduced from 92.5% to 11.0%. Originality: the paper does an excellent job of describing the related literature and enumerating its contributions that go beyond that literature. While previous work has performed adversarial attacks against ASR systems and incorporated robustness to reverberation into it, this appears to be the first to work against a commercial system over the air. Quality and clarity: the paper is well written and clearly describes the necessary pieces and tools used to create the system and test it. The experiments are thorough, testing the attack digitally and physically on the emulated system and physically on the Echo. The attack was also tested with different relative positioning to the Echo (distance of the attacker, the talkers, and angular separation between the two). One aspect that is not clear is the duration of the melody and its temporal relationship to the speech. How many notes long is it? Does it just loop after that? Does it matter when it occurs relative to "Alexa"? It is rather surprising that such a relatively simple attack could work so well. While recorded guitar music played at the same volume is a good baseline for comparison, a better baseline would be using the same Karplus-Strong synthesis with different frequencies and intensities showing that they do not interfere. Perhaps an optimization in the opposite direction (maximize intensity while minimizing detection disruption) would show how much contrast is achievable, i.e., how much of the attack comes from some sound being played while the person is speaking vs this particular sound being played. Significance: the significance of this work is high. It appears to be the first adversarial attack against a commercial ASR system and it works over the air. As such, it is likely to be cited by many researchers going forward. One aspect that diminishes the significance of the work is that its effect is only to prevent wake-word detection and not to enable it falsely or convert one utterance into another. Thus its potential for actual attacks is relatively small, but it does present a methodology that could be used in more serious attacks going forward. General comments: Since the attack is basically only controlling the frequency and intensity of notes in a melody, it would be interesting to know how crucial the Karplus-Strong algorithm is here. Could a MIDI synthesizer be used instead with the same parameters and a pluck-like patch? What about other instruments? Experiments using this same method to activate the wake-word detection instead of deactivating it would be very interesting. Minor comments ---------------------- Lines 33 and 42: "It is revealed that..." who or what has revealed these pieces of information about the Echo? If a paper, please cite it. Line 52 and 253: "varying volume, tempo, and speed" of utterances. What is the difference between tempo and speed? They seem to be redundant here. Perhaps something else is meant? Rhythm? Line 168: frequency of a note is not the key of a piece. If "key" is meant as in key of a piano, this is confusing. Please remove "(key)". Line 198: 'compress the loose-less file format "wav" to the loosely file format "mp3"' These should be "lossless" and "lossy". Line 208: the square bracket is not used in (3), perhaps another symbol was meant? Table 2: This would be clearer if results were reported as decimal values with N=10 in the caption. Author feedback --------------------- I have read the author feedback and appreciate the additional experiment using the karplus-strong algorithm to generate comparable non-adversarial examples that do not provide any ability to disrupt wake-word recognition. These will further strengthen the paper, as will experiments on the synchronization of the wake word with the music.

Reviewer 3



The paper presents a successful audio attack on a commercial grade Automatic Speech Recognition system - Alexa. To my best knowledge it is the first time it was successfully performed, and so claim the authors. What is worth emphasizing is that the attack is performed fully in the real world. An actual piece of music is recorded and played using actual speakers. It interferes with the word "Alexa" spoken by a human to a physical Alexa device. The device is placed in a reasonable distance from the speaker and the distracting device in a physical room. The authors provided an actual video recording of the experiment taking place, which I find very valuable and working very much in favor of the acceptance of the paper for NeurIPS 2019. The authors claim the attack was a black box attack, but they relied heavily on the works of Panchapagesan, Gao and others to work out the adversarial signal, leveraging the work of those on "reverse engineering" Alexa's algorithm. The authors themselves admit, the gradients of a real Alexa can be expected to be similar to those of models of Panchapagesan and Gao, so I would it call it a "gray box" attack. What I find most valuable about the paper is: 1) It works. Plain and simple - the adversarial music played by the authors in the real life setting makes Alexa not wake up to "Alexa" command spoken clearly and audibly. 2) Video is provided, and it demonstrates that a physical meter was used to measure the levels of audio signal and the voice of the speaker, contributing to the credibility of the result. The task of building real-life attack on Alexa was nontrivial, and I like the approach presented. Instead of reinventing the wheel the authors cleverly leveraged the existing results to come up with an efficient pipeline that lead to the desired outcome. I find the discussion provided by the authors not entirely convincing. The adversarial signal does not sound that musical to me (and I happen to be a progressive-metal guitar player). This is subjective, of course, but I do not think that many people would be deliberately listening to that type of sounds for entertainment. It was generated from a synthesized guitar track (why not a real guitar?), yet it sounds more like surrealistic steel drums/handpan to me, so the adversarial perturbation was very significant. The transients are really boosted here. Also, the perturbing signal is quite loud. mp3-s are mentioned, and indeed, lossy compression is very relevant here. The whole goal of the lossy compression is to leverage psychoacoustic effects and "hide" the compromises from the listener. It would be really interesting, if the adversarial signal could be made sound more pleasing to human ears using similar techniques. Maybe some mp3-inspired constraints on Projected Gradient Descent? That could be a fascinating theoretical research topic. Overall however I find the experiment very valuable and I am recommending the paper for publication at NeurIPS 2019.

[Author Response · NeurIPS 2019]

| Test Against Alexa | $\phi = 0°$ | | |
|---|---|---|---|
| $d_t =$ | $4.2ft$ | $7.2ft$ | $10.2ft$ |
| Random Music generated by the Karplus-Strong(KS) algorithm | 0/10 | 0/10 | 0/10 |
| Random Notes generated by the Karplus-Strong(KS) (single note) algorithm | 1/10 | 0/10 | 0/10 |

Table 1: Times of the real Amazon Alexa being able to respond to the wake-word under the influence of baseline noises.

Thanks to all the reviewers for your time and detailed comments! For the most part, we fully agree with many of the statements that the reviewers make about this paper. As was recognized by all, this paper is about introducing a denial-of-service (DoS) attack against voice assistant systems, and as such its motivation and value lies largely in the evident fact that such attacks are possible in the real world against a commercial-grade product, rather than in the algorithmic components.

-Reviewer #1 Thank you for your suggestion on problem setup, and we will modify our problem statement accordingly to focus on DoS attack (false-negative). To answer your question about the baseline, we experimented with two new sample audio generated by the same (Karplus-Strong) algorithm and tested against Alexa. The result is shown in Table.1. The musical audio does not fool Alexa. (Please see another demo video here: `https://youtu.be/TRHGpYzv_Sk`, uploaded anonymously. Though we understand if the reviewers are not able to take it into account for review, as the response guidelines mentioned not to reference external links.) As for the timing offset, we have been conducting a more rigorous quantitative analysis. If we were given a chance to present in the camera-ready version in the main conference, we would definitely include a thorough comparison with the different length of timing offset from alignment. So far, from our rough empirical evaluation and as you can see in the original demo video, we observe that our adversarial music does not have to be aligned with the wake word perfectly, but the audio tends to work the best while being looped. If the wake word starts first, it would be recognized by Alexa. If the noise starts first, the wake word is nullified. We will also include a digital experiment in our camera-ready version with specific digital simulation evaluating the performance of the model as a function of SNR. We will also add more physical experiments to study the trend of SNR VS accuracy in the real world. Thank you again for your constructive feedback!

-Reviewer #3 The Karplus-Strong (KS) algorithm is used here since the formulation, and implementation of the algorithm is fully open-sourced and differentiable with current configuration (controlling the frequency and the intensity of notes), which could be plugged nicely into our adversarial training paradigm. We expect other MIDI synthesizers could be effective given the right configurations as well. If we were given a chance to present our work at the main conference, we would add our result of our on-going effort trying to generate our adversarial music using Google Magenta Synthesizers Nsynth. As for your question about the baseline, please see our answer to Reviewer #1 and the additional demo video.

Currently, we are also trying to activate the wake-word using our adversary. So far, we observe false positives are way more challenging to produce than false-negatives if we were using the same music format in the same adversarial training paradigm. However, a distortion of the recorded wake-word could easily trigger the detection of false-positives, which makes false positive less attractive in the scope of our discussion. Thank you again for your encouragements and constructive feedback!

-Reviewer #4 Thank you again for your constructive feedback! We agree with your suggestion calling our model "gray box" attack. Please see our answer to R#3 for the motivation behind the KS algorithm. We conjecture that the simplicity of the KS algorithm also constrains us since the parameters and complexity heavily constrain the variability of our adversaries that the KS algorithm defines. This could be the main reason for the transient-heavy and high-volume sound. Currently, we are working on other synthesizers and other instruments to make our adversary sound more pleasing to humans. As of now, we observe guitar works the best. A more complex synthesizer (e.g., a neural-network-based synthesizer Nsynth) might be able to provide us more attacking budgets due to more degree of freedom with a lot more parameters in constructing the adversary. (even though itself is a black box) If given the opportunity for the camera-ready submission, we will try our best to provide more low-volume low-attack adversaries and explore the SNR threshold for our adversary to be effective as you suggested. Although from what we have observed, the universal VA adversary could be very difficult if not impossible to achieve in the real world, since the adversary itself needs to be robust against other environmental noise.



[Meta-Review · NeurIPS 2019]

This is an important paper, that presents a successful adversarial audio attack on a (well-emulated) commercial-grade voice assistant.